# Molecular and Cellular Mechanisms of Action of Cannabidiol

**DOI:** 10.3390/molecules28165980

**Published:** 2023-08-09

**Authors:** Nadia Martinez Naya, Jazmin Kelly, Giuliana Corna, Michele Golino, Antonio Abbate, Stefano Toldo

**Affiliations:** 1Robert M. Berne Cardiovascular Research Center, Division of Cardiovascular Medicine, School of Medicine, University of Virginia, Charlottesville, VA 22903, USA; zfw3js@virginia.edu (N.M.N.); jas5rc@virginia.edu (J.K.); tyz2qs@uvahealth.org (A.A.); 2Pauley Heart Center, Division of Cardiology, Department of Internal Medicine, Virginia Commonwealth University, Richmond, VA 22903, USA; giuliana.corna@vcuhealth.org (G.C.); michele.golino@vcuhealth.org (M.G.); 3Interventional Cardiology Department, Hospital Italiano de Buenos Aires, Buenos Aires 1199, Argentina; 4Department of Medicine and Surgery, University of Insubria, 2110 Varese, Italy

**Keywords:** cannabidiol, CBD, inflammation, mechanisms, molecular target

## Abstract

Cannabidiol (CBD) is the primary non-psychoactive chemical from Cannabis Sativa, a plant used for centuries for both recreational and medicinal purposes. CBD lacks the psychotropic effects of Δ9-tetrahydrocannabinol (Δ9-THC) and has shown great therapeutic potential. CBD exerts a wide spectrum of effects at a molecular, cellular, and organ level, affecting inflammation, oxidative damage, cell survival, pain, vasodilation, and excitability, among others, modifying many physiological and pathophysiological processes. There is evidence that CBD may be effective in treating several human disorders, like anxiety, chronic pain, psychiatric pathologies, cardiovascular diseases, and even cancer. Multiple cellular and pre-clinical studies using animal models of disease and several human trials have shown that CBD has an overall safe profile. In this review article, we summarize the pharmacokinetics data, the putative mechanisms of action of CBD, and the physiological effects reported in pre-clinical studies to give a comprehensive list of the findings and major effects attributed to this compound.

## 1. Introduction

Cannabis Sativa, also known as marijuana or hemp, is an indigenous plant from Eastern Asia, which synthesizes several chemical compounds. To date, 554 chemical compounds have been identified, including 113 phytocannabinoids [1,2] and 120 terpenes, which are responsible for its characteristic aroma [2,3].

Marijuana was the most-consumed illicit drug in United States in 2019 and has been used for several centuries, mainly in a recreational fashion. It exhibits a wide range of medicinal properties, some of which are desired, such as analgesia, anti-inflammation, immunosuppression, anti-convulsant properties, and attenuation of vomiting. However, prejudicial consequences have also been described, since it has been associated with impaired cognition with long-lasting effects, increased angina frequency, changes in sympathetic and parasympathetic nervous system signal transduction, central and peripheral vasoconstriction, Raynaud’s phenomenon, ischemic ulcers, and hypertension, among others [3].

Δ9-tetrahydrocannabinol (Δ9-THC) is the main psychoactive compound of cannabis, while cannabidiol (CBD) is the primary non-psychoactive one [2,4].

CBD was first isolated from Mexican marijuana by Adams et al. [5] in the late 1930s. Its structure was elucidated in 1963 by Mechoulam and Shvo after extraction from Lebanese marijuana, a concentrated product made of purified cannabis preparations [6,7].

### 1.1. CBD as a Therapeutic Option

In the United States of America, the Food and Drug Administration (FDA) has approved the use of a highly concentrated plant-based CDB preparation in the treatment of seizures associated with Dravet and Lennox–Gastaut syndromes and tuberous sclerosis complex [8].

CBD can also be produced synthetically, yielding a pure form of CBD [9,10]. The efficacy of the plant-based CBD purified formulation versus the synthetic form has been shown to be similar regarding pharmacokinetics and effect. Regardless, natural CBD formulations can contain other phytocannabinoids that could be present at moderate concentrations, modifying its effects or adverse reactions [9].

Over the last few years, cannabinoid chemistry and pharmacology have been the object of thousands of publications. Basic and/or clinical studies have shown that CBD has multidirectional properties and uses. However, most of these findings require further investigation to confirm clinical effectiveness in human beings.

The aim of this review article is to summarize the available information on pharmacology and pharmacokinetics and to list the different biological effects of CBD in cellular and animal models.

### 1.2. Chemical Structure

The term ‘cannabinoid’ refers to chemical substances with a characteristic twenty-one carbon-atom terpenophenolic structure which has the ability to interact with the cannabinoid receptors. There are two distinct groups of compounds that engage with the endocannabinoid system (ECS) of vertebrates: endocannabinoids, which are endogenous lipid-based neurotransmitters within mammalian bodies, and phytocannabinoids, which are naturally occurring compounds found in the cannabis plant. From a chemical perspective, the endogenous ligands of endocannabinoids primarily consist of eicosanoid compounds synthesized in the lipid bilayer. They are rapidly synthesized and metabolized, resulting in a relatively short duration of action. They act “on demand” to uphold homeostasis and regulate various physiological processes, like pain, mood, appetite, sleep, and immune response [1,2,3]. The structure of phytocannabinoids differs significantly from that of endocannabinoids. The duration of their effects can be prolonged depending on the method of consumption and the individual’s metabolism. In addition, although phytocannabinoids interact with the ECS, they can also exhibit other effects independent of their activity on the ECS. This review comprehensively outlines the relevant mechanisms of cannabidiol (CBD), some of which overlap with endocannabinoids, while others are independent [1,2,3].

CBD shares the exact chemical formula of THC, C21H30O2 [10,11]. However, structurally, there is one main difference: whereas THC contains a cyclic ring, CBD contains a hydroxyl group (Figure 1) [6,12]. From this point of view, the saturated exocyclic C−C double bond provides no pathway for the conversion of CBD to psychoactive Δ9-THC.

The slight difference in molecular structure between these two compounds confers profoundly different pharmacological properties [1,2] (Figure 1).

## 2. Pharmacokinetics

CBD is a small, lipophilic molecule; therefore, it is sequestered in fatty tissues and penetrates highly vascularized tissue (such as adipose, heart, brain, liver, lungs, and spleen) with subsequent equilibration into less-vascularized tissue and a rapid decrease in plasma concentration [13,14]. For this reason, the absorption, distribution, metabolism, effects, and elimination of CBD are highly affected by its pharmaceutical formulation, route of administration, dosage schedule (single vs. multiple doses), and diet [15]. CBD can be ingested as lipid/oil-based formulations, gelatin matrix pellets, and self-emulsifying drug delivery systems (SEDDS) [15,16]; it can also be vaporized and delivered through intravenous or transdermal administration. CBD has a protein-binding capability of >95%, mainly to lipoproteins [12], and has a long terminal elimination half-life of 56 to 61 h after twice-daily dosing for 7 days [17]. Following a single intravenous dose, its half-life has been observed to be 24 ± 6 h, and post-inhalation to be 31 ± 4 h [8,18]. CBD induces rapid liver metabolization through an extensive first pass extraction, after which its metabolites are excreted through the fecal route and, to a lesser extent, urinary excretion [14]. Controlled human studies to test CBD have predominantly utilized oral administration. However, CBD’s oral bioavailability is limited and is estimated to be as low as 6% [1,2]. Existing data from orally administered CBD formulations indicate good tolerance, with only a few mild-to-moderate adverse events strongly related to dosage. Information on the differential rates of CBD absorption through various application routes, such as smoking, vaporization, oral soft gels, and oil drops, is scarce, inconsistent, and results in highly variable pharmacokinetic profiles. Therefore, predicting the appropriate dose and route of administration to enhance desired effects and minimize adverse consequences becomes challenging in the absence of enough clinical data supporting the safety and efficacy of these compounds.

### Safety and Adverse Effects

CBD is considered a safe drug. Its use is associated with a few mild/moderate adverse events which are strongly related to the dosage [19]. Serious adverse events are rare (reported from 3% to 10%), and include elevation of alanine aminotransferase (ALT) and/or aspartate aminotransferase (AST), pyrexia, upper respiratory tract infection, and convulsions [20]. Increases in transaminases were mainly reported in clinical trials involving epileptic patients and were explained by the ability of CBD to inhibit the hepatic metabolism of other drugs (e.g., clobazam and sodium valproate) [19].

## 3. The Endocannabinoid System

### 3.1. Cannabinoid Receptors and Endocannabinoids

The first cannabinoid receptor, CB1, was isolated as a G protein-coupled receptor derived from rat cerebral cortex cDNA mediating pharmacological effects of Δ9 -THC in 1988 [21,22]. In 1990, a protein homologous to CB1 and with an affinity for Δ9 -THC was identified and was called CB2 [23]. Initially, CB2 expression could only be detected in rat spleens, being especially present in marginal zones around the periarteriolar lymphoid sheets, and was later identified as located in the macrophages/monocytes population. However, CB2 was also later found in the brain [24,25], in the endocrine pancreas [26], and in the bones [27,28,29]. It became clear that the peripheral effects of cannabinoids were mediated by CB2, and that it could have a possible role in inflammatory and immune responses [30,31,32].

‘Endocannabinoids’ are endogenous lipidic agonists for these receptors, and the main ones are arachidonoyl ethanolamide (Anandamide), 3,2-arachidonoyl glycerol, and 4–6 and 2-arachidonyl glyceryl ether (Noladin ether) [33,34]. All of these, together with cannabinoid receptors CB1 and CB2, comprise the ‘endocannabinoid system’ [2].

Endocannabinoids have been extensively studied, and their biosynthesis, cellular transport, metabolism, and biological function have guided CBD research.

CBD acts through multiple mechanisms that largely have been investigated but are not fully understood. Selective CB2 agonists have been synthesized to avoid unwanted psychotropic effects associated with CB1 activation. Despite these efforts, the clinical outcomes using these CB2 ligands have shown limited effectiveness [35].

### 3.2. CBD and the Cannabinoid Receptors

CBD has a low binding affinity for CB1 and CB2 [23,36,37,38]. Nevertheless, some in vivo effects of CBD seem to be dependent on the presence of the CB1 receptor. CBD increases adult neurogenesis through CB1, since CBD has been shown to have no effect in CB1 knockout animals [39]. CBD displays an unexpected high potency as a non-competitive antagonist of CB1 and CB2 in the mouse vas deferens and brain [30,40], and can behave as a CB2 receptor inverse agonist [40]. This action has been described as a non-competitive negative allosteric modulator, also reducing the intracellular G-dependent signaling. CBD binds to an allosteric site on CB1 receptors that is functionally different from the orthosteric site necessary for endocannabinoid signaling [35].

Therefore, CBD is not a primary ligand of CB1 or CB2, but it may influence their signaling by modifying endocannabinoid tone [35]. The use of CBD over specific CB2 agonists offers several advantages. CBDs can target multiple pathways associated with metabolic and inflammatory processes in the cell. Moreover, CBD’s anti-inflammatory effects extend beyond CB2 receptor activation. While specific CB2 agonists may have their advantages in certain cases, such as in targeted immune modulation, using CBD may offer a more comprehensive and versatile approach to address multiple health conditions with fewer potential drawbacks. As research on CBD and its effects continues to expand, its therapeutic potential and utility in various medical contexts are becoming increasingly evident.

## 4. CBD Effects on Membrane Receptors

### 4.1. Transient Receptor Potential Cation Channels (TRP)

TRP channels actively participate in signal transmission, triggered by a wide range of chemical and physical stimuli, including intense heat or acidic environments. These channels are a family of trans-membrane ion channels on sensitive peripheral nerves. Activation of TRP channels is characterized by a two-phase action, with an excitatory phase characterized by pain and/or neurogenic inflammation followed by a lasting refractory state commonly referred to as desensitization [41].

CBD interacts with six TRP channels, activating TRP V1 (Vanilloid or VR1), TRP V2, TRP V3, TRP V4, and TRP A1 (Ankyrin), and antagonizing TRP M8 (Mucolipin) [42,43].

The vanilloid receptor type 1 (TRPV1), via the release of inflammatory and algetic peptides, is involved in inflammatory hyperalgesia [44]. When TRPV1 is stimulated by capsaicin and certain analogs, it undergoes rapid desensitization, leading to unexpected paradoxical analgesic and anti-inflammatory effects [41]. TRPV1 and TRPV2 transduce inflammatory and chronic pain signals at both peripheral and spinal levels. Some of CBD’s anti-hyperalgesic properties can be explained by interactions with these channels.

CBD is a weak agonist of human TRPV1 and lowers their sensitivity to capsaicin, thus leading to the possibility that this cannabinoid exerts anti-inflammatory action in part by desensitization of sensory nociceptors [45].

Another relevant mechanism of CBD is enhancing endocannabinoid actions. The chemical similarity between anandamide and olvanil suggests that certain vanilloids might interact with either of the two cannabinoid receptors, or with the anandamide transporter or the fatty-acid amide hydrolase (FAAH). Anandamide exerts anti-inflammatory and neuroprotective actions. CBD inhibits anandamide amidase [46], is responsible for its enzymatic hydrolysis, and inhibits transporter-mediated anandamide uptake by cells, thereby enhancing the putative tonic inhibitory action of anandamide on inflammation [45] and favoring desensitization of sensitive neurons (Figure 2).

### 4.2. Serotonin Receptor 1 A

Serotonin receptor 1A (5-HT1A) plays a critical role in the pathophysiology of depression, aggression, and anxiety. CBD is an agonist of 5HT1A with a micromolar affinity and might exert anxiolytic effects by activating its post-synaptic interaction [47,48]. In this receptor, the glutamate pyruvate transaminase (GTP)-binding proteins are responsible for the coupling between 5-HT1A activation and its subsequent effect, and CBD has demonstrated the ability to increase GTP binding to the receptor-coupled G-protein, Gi, which is a characteristic behavior of a receptor agonist [48]. CBD also increases serotoninergic and glutamatergic transmission through a positive allosteric modulation of 5-HT1A [20,38] (Figure 2).

### 4.3. GABAA Receptors

Gamma-aminobutyric acid (GABA) plays a key role in vertebrates’ central nervous systems, facilitating a rapid inhibitory neurotransmission, and causing brain hyperexcitability through its interactions with GABAA receptors. CBD is an allosteric modulator of GABAA receptors, and amplifies the currents produced by low, but not by high, GABA concentrations, effectively increasing GABA’s apparent affinity for its receptor [49].

Due to the regulation of GABA signaling, CBD could interfere in different pathophysiological brain processes such as anxiety and sleeping disorders and may reduce seizures.

Interestingly, the effects of CBD on GABAA are easily reversible, indicating that its action does not rely on intracellular pathways [50] (Figure 2).

### 4.4. Nuclear Peroxisome Proliferator-Activated Receptors (PPAR)

PPARγ is a key transcription factor regulating adipocyte differentiation [51] and lipid and glucose homeostasis [52]. In addition to its role in metabolic tissues, some of the beneficial effects of PPARγ ligands are due to anti-inflammatory actions, including inhibition of pro-inflammatory cytokines such as interferon gamma (IFNγ) and tumor necrosis factor-alpha (TNFα) [53,54], increase of anti-inflammatory cytokines, and inhibition of inducible nitric oxide synthase (iNOS) expression. It is expressed in several immune-cell types, such as macrophages, dendritic cells, T cells, and B cells [53], and affects nuclear factor kappa beta (NF-κB) transcriptional activity by inhibiting the inhibitor of κB (IκB) kinase [55,56] and the DNA binding domains of NF-κB [53,57].

Endocannabinoids, endocannabinoid-like compounds, phytocannabinoids, and synthetic cannabinoids bind and activate PPARs [58,59]. CBD binds to, and increases the transcriptional activity of, PPARγ, mediating its effects both in the vasculature and in adipocytes [60].

CBD, via PPARγ, increases anti-inflammatory cytokines, inhibits iNOS expression, and decreases the inflammatory response in cardiovascular cells, particularly endothelial cells. Furthermore, CBD decreases monocyte adhesion and trans-endothelial migration [9,61,62] and reduces the expression of the adhesion molecule VCAM in human brain microvascular endothelial cells [63,64]. CBD also protects against β-amyloid neurotoxicity and inflammation in rats through PPARγ agonism [61,65].

In summary, the evidence suggests that PPARγ is a key factor in CBD’s ability to suppress the inflammatory response (Figure 2).

### 4.5. Adenosine Receptors

Adenosine release is an endogenous mechanism evoked during cellular stress or inflammatory activation and it mediates an autoregulatory loop by which immunosuppression protects the organs from the injury caused by the initiating stimuli [66].

Adenosine prevents the activation of IκB kinase subunit β (IKK-β) and NF-κB translocation to the nucleus [67]. It inhibits adhesion and cytokine release of stimulated neutrophil [67], and, in monocytes, it binds the adenosine A2 (A2A) receptors and inhibits TNF-α [68,69], IL-6 [70], and IL-12 [71], while it enhances IL-10 synthesis [72,73].

CBD exerts some of its effects through the activation of adenosine receptors A1 and A2, which has raised the possibility that CBD might act as an adenosine receptor agonist. Consequently, several trials have proved that the co-treatment of CBD and A1 or A2 adenosine receptor blockers abolishes the ability of CBD to elicit a response through adenosine receptors [74,75,76]. The literature suggests that direct agonism of A1/A2 receptors by CBD is unlikely due to its concentration–effect (E/c) curve and surplus agonist concentration, and, as such, indirect elevation of local adenosine levels is not only more plausible, but could represent a viable mechanism underlying CBD’s actions through those receptors [73,77,78,79].

CBD indisputably affects the adenosinergic signaling through a competitive inhibition of adenosine uptake at the equilibrative nucleoside transporter 1 (ENT1), as evidenced in neurons, macrophages, and retinal and brain microglial cells, and in the myocardium [77,78,79,80], it increases its endogenous activity.

This adenosine signaling enhancement by CBD in vivo activates the A1 receptor and can exert an antiarrhythmic effect during ischemia/reperfusion [74], while the A2A receptor activation is responsible for some of the drug’s observed anti-inflammatory effects, like the decrease in serum TNFα, IL-6, and cyclooxygenase-2 (COX-2) and iNOS expression after treatment with lipopolysaccharide (LPS) [77,78]. Also, the drug is responsible for the transmigration of blood leukocytes by downregulating the expression of vascular cell adhesion molecule-1 (VCAM-1), chemokines (CCL2 and CCL5), and the pro-inflammatory cytokine IL-1β. It also attenuates the activation of microglia in a mouse model of multiple sclerosis [81] (Figure 2).

## 5. CBD’s Effect on Inflammatory Signaling

### 5.1. NF-κB and Interferon Beta

The dual nature of the inflammatory response has been broadly described. While, in the short term, it exerts a protective role against infections and injuries, persistent or chronic activation potentially leads to adverse consequences and participates in the development of various chronic conditions.

CBD modulates the function of the cells of the immune system and exerts anti-inflammatory and antioxidant effects, and its anti-inflammatory effects can be attributed to the NF-κB and the interferon beta (IFN-β) pathways [82].

#### 5.1.1. NF-κB

The NF-κB pathway includes a family of inducible structurally related transcription factors, including NF-κB1 (p50), NF-κB2 (p52), RelA (p65), RelB, and c-Rel. These mediate the transcription of target genes by binding to a specific DNA element, κB enhancer [82].

The inhibitors of the κB (IκB) family sequester NF-κB proteins in the cytoplasm, preventing the activity of these transcription factors in the absence of pro-inflammatory or stress signaling [83]. When activated, this pathway orchestrates the transcription of numerous inflammatory genes, including those encoding TNF-α, IL-1β, IL-6, IL-12p40, and COX-2 [84]. LPS activation of toll-like receptor (TLR)- 4 leads to IκB inactivation via interleukin-1 receptor-associated kinase 1(IRAK-1) -dependent phosphorylation, which is followed by ubiquitin-dependent degradation, leading to p65 nuclear translocation. Moreover, canonical NF-κB members RelA and c-Rel have a significant role in mediating TCR signaling and naive T cell activation [84].

CBD decreases IRAK-1 degradation and reverses the IkB degradation, ultimately reducing NF-κB p65 nuclear translocation [84,85]. CBD also influences the interaction of transcription factors like nuclear-factor-erythroid-2-related factor 2 (Nrf2) with NF-κB, increasing the expression of Nrf2 activators, and stimulating the transcription activity of Nrf2, inhibiting the NF-κB pathway [4]. CBD also suppresses NF-κB-mediated transcription by increasing anti-inflammatory STAT3 phosphorylation while reducing pro-inflammatory STAT1 phosphorylation [84].

Furthermore, IL-1 β, essential for the host response and resistance to pathogens, leads to the activation of NF-κB [86]. Since CBD also reduces IL-1β synthesis, it prevents NF-κB activation downstream of this pathway [87]. CBD is a potent inhibitor of the NF-κB pathway, but the exact mechanism or molecular target by which CBD reduces NF-κB signaling is unknown (Figure 3).

#### 5.1.2. IFN-β

Interferon regulating factor 3 (IRF-3) binds the IFN-stimulated response-elements-DNA sequence, inducing the production of the IFN-β cytokine [84]. IFN-β activates a second wave of gene expression, mostly chemokines, such as interferon-γ-inducible protein 10 kDa (CXCL10), C-C motif ligand 5 (CCL5), and C-C motif ligand 2 (CCL2), by binding to an IFN receptor and inducing phosphorylation of Janus kinase (JAK), leading to the signal transducer and activator of transcription proteins (STAT) pathway activation [84]. CBD (10 μM) inhibits IFN-β transcription and synthesis [88] by inhibiting interferon regulating factor 3 (IRF-3) [86]. Although the exact mechanism of action is unknown, it is likely that CBD targets the upstream phosphorylation, and therefore the nuclear sequestration, of IRF-3. CBD alters STAT1 phosphorylation following LPS treatment, suggesting that JAK-STAT signaling may mediate the mechanism by which CBD regulates IFN-β-dependent inflammatory processes in peripheral blood mononuclear cells [82,89]. As for the NF-κB signaling cascade, the exact mechanism or molecular target by which CBD regulates IFN-β is unknown.

Collectively, these studies support the theory of immunomodulatory properties of CBD due to the downregulation of the NF-κB pathway and modulation of the IFN-β/STAT signaling cascade [84].

### 5.2. NLRP3 Inflammasome

The NACHT-, LRR- and pyrin-domain-containing protein 3 (NLRP3) is an apical proinflammatory receptor controlling the innate immune response. It detects pathogen-associated molecular patterns (PAMPs) from microbes and danger-associated molecular patterns (DAMPs) from host-derived damaged cellular and extracellular material linked to sterile inflammation. Upon sensing the initial pro-inflammatory signal, NF-κB triggers non-transcriptional or transcriptional NLRP3 inflammasome priming, leading to increased expression of the NLRP3 inflammasome pathway proteins. Persistent pro-inflammatory and stress signaling promote NLRP3 activation and assembly of the inflammasome, a multiprotein complex containing the apoptosis speck-like protein containing a caspase recruitment domain (ASC) and pro-caspase-1 [90]. Inflammasome formation induces caspase-1 activation, an enzyme necessary for the conversion of pro-interleukin-1β (pro-IL-1β) and pro-IL-18 into their mature forms [91].

There are a few recent studies that have investigated the effect of CBD on NLRP3 activation. Huang et al. [92] showed that CBD significantly inhibits NF-κB p65 nuclear translocation and the activation of NLRP3 inflammasome, both in vivo and in vitro studies, in a liver inflammation induced by high-fat high-cholesterol diet model, which leads to the reduction of the expression of inflammation-related factors. In human gingival mesenchymal stem cells, CBD prevents NLRP3-inflammasome pathway activation by suppressing the expression of key genes, including NLRP3 and caspase 1, and inhibiting downstream production of IL-18. CBD has induced down-regulation of pro-inflammatory cytokines and genes associated with the IL-1 pathway [93].

Moreover, LPS was used to significantly increase the production of IL-1β in monocytes, and CBD successfully attenuates this IL-1β production [84,94]. Furthermore, CBD in concentrations of 0.1, 1, and 10 μM inhibits the NLRP3 inflammasome activity through reduced expression of NLRP3 and IL-1β mRNA, which is associated with reduced IL-1β secretion in vitro [87].

Overall, the data support the concept that CBD inhibits NLRP3 inflammasome activation via the inhibition of NF-κB, reducing inflammasome priming [84,94,95].

### 5.3. IFN-γ

IFN-γ plays a pivotal role in the host defense system. It is a key regulator of type 1 T helper (Th1) lymphocytes, CD8 lymphocytes, B cells, natural killer (NK) T cells (NKT), and antigen-presenting cells like monocytes, macrophages, and dendritic cells. This cytokine is responsible for orchestrating both innate and adaptive immune responses. Notably, IFN-γ enhances the capacity of cytotoxic T cells to identify foreign peptides, thereby promoting the development of cell-mediated immunity [96,97].

CBD treatment significantly reduces plasma levels of proinflammatory cytokines (IFN-γ, TNF-α) produced by activated Th1 cells and peritoneal macrophages and prevents the onset of autoimmune diabetes in non-obese diabetes (NOD)-prone mice [98,99]. A possible mechanism of CBD-mediated suppression of IFN-γ is associated with the suppression of the transcriptional activity of activator protein-1 (AP-1) and nuclear factor of activated T cells (NFAT) [100], raising the question of whether CBD may target a common protein upstream of AP-1, NFAT, NF-κB, and IFN3, or whether these are independent targets.

### 5.4. TNF-α

The administration of CBD leads to the suppression of T cell response and reduces TNF-α release from synovial cells obtained from arthritic knee joints in mice [101,102]. Moreover, a single CBD dose resulted in decreased serum TNF-α levels in mice treated with lipopolysaccharide (LPS) as a consequence of the activation of the A2A adenosine receptors [6,77]. CBD also alleviated the TNF-α-mediated expression of pro-inflammatory cytokines IL-1β and IL-6 [103].

CBD treatment significantly reduces the expression of M1 macrophage-related genes (TNFα and MCP-1), suggesting that inhibition of M1 polarization could contribute to the anti-inflammatory effects of CBD [104,105,106,107]. Suppression of TNF-α expression and effects may be a direct effect of CBD inhibition of NF-κB signaling.

### 5.5. Oxidative Damage

Reactive oxygen species (ROS) are bioproducts of the normal metabolism of oxygen that serve as key regulators of several functions. Immune cells produce ROS via the NADPH oxidase 2 (NOX2) complex as a mechanism to eradicate pathogens [108]. ROS production is both a consequence and a cause of inflammation. Enhanced levels of cytokines or the presence of PAMPs and DAMPs induce ROS. Conversely, ROS cause irreversible damage to DNA, lipid peroxidation, and enzyme inactivation, and when persistent, can ultimately lead to cell death and tissue destruction [109,110,111]. Furthermore, nitric oxide (NO), which can originate locally or from cells that infiltrate the site of inflammation [112], rapidly reacts with free radicals, namely, superoxide anions, inducing lipid peroxidation and promoting generalized oxidative/nitrosative damage [109,113]. ROS promote activation of the NLRP3 inflammasome during inflammation [114,115] and contribute to NF-κB signaling.

CBD possesses intrinsic antioxidant effects [116,117], since it has the ability to donate electrons, being oxidized in the process [118]. CBD inhibits mitochondrial superoxide generation in high-glucose-stimulated human coronary endothelial cells and diabetic mice by indirectly increasing the iNOS expression and 3-NT formation [105,119] and, therefore, attenuating mitochondrial ROS generation and simultaneously reversing the abnormal changes in antioxidant biomarkers following hippocampal oxidative damage post-oxygen–glucose-deprivation/reperfusion injury.

Experimentally, the use of CBD can also attenuate xanthine oxidase (XO) activity in keratinocytes exposed to UVB irradiation and H_2_O_2_ [11] and reduces the expression of the superoxide generators RENOX (NOX4) and NOX1 in a mouse model of cisplatin-induced nephrotoxicity [109,120]. Repeated doses of CBD suppress the lipid peroxide overproduction in paw tissue of neuropathic and inflamed rats [121]. CBD can also increase the activity of multiple antioxidant enzymes like superoxide dismutase (SOD), glutathione peroxidase-1 (GPx-1), glutathione (GSH), and glutathione peroxidase [105,122,123].

Furthermore, CBD had indirect effects on mitochondrial function, improving basal mitochondrial respiration and the rate of ATP-linked oxygen consumption, as well as increasing glucose consumption in a neuronal cell line model of oxygen/glucose deprivation and reperfusion. Additionally, the activation of glucose 6-phosphate dehydrogenase and the preserved NADPH/NADP+ ratio indicate that CBD stimulates the pentose phosphate pathway [104,122]. It also prevents oxidative stress generated by microglial cells in response to LPS exposure, probably operating by inhibition of ROS-dependent activation of NF-κB [116], and regulates redox-sensitive transcription factors such as Nrf2; a key role of Nrf2 is initiating the transcription of antioxidant and cytoprotective genes in microglia [124].

CBD also protects against vascular damage by attenuating oxidative/nitrative stress, inflammation, cell death, and fibrosis in the high-glucose environment of a rat model of type 2 diabetes [105].

Nevertheless, it is difficult to point out which effects of CBD are due to its antioxidant properties, anti-inflammatory activity, or a direct effect on specific enzymes/proteins involved in the regulation of ROS.

## 6. CBD Modulates Inflammatory Cell Functions

### 6.1. Neutrophil Activation

The neutrophil recruitment cascade to inflamed tissues involves neutrophil rolling and adhesion to the activated endothelium, extravasation by chemotaxis through breach of the endothelial barrier, recognition of activating signal, phagocytosis, and occasionally, release of neutrophil extracellular traps (NETs).

Experimentally, CBD reduces chemotactic mediators, mainly consisting of inflammatory cytokines like TNF-α, IL-1β, and IL-8, reducing the chemotaxis of neutrophils [125,126,127].

CBD interferes with the translocation of the NOX2 subunits to the membrane, preventing the oxidative burst and, consequently, the generation of O2− and H2O2 [104]. Additionally, CBD reduces neutrophil activation and degranulation [128].

CBD reduces the hepatic expression of E-selectin/CD62, or endothelial-leukocyte adhesion molecule 1, which is a key adhesion factor expressed in the activated endothelium, which is involved in the recruitment of leukocytes, particularly neutrophils [119,129].

Furthermore, CBD attenuates high-glucose-induced upregulation of adhesion molecules ICAM-1 and VCAM-1, trans-endothelial migration of leukocytes, leukocyte-endothelial adhesion, and disruption of the endothelial barrier function in human coronary arteries in a dose-dependent manner [119]. Also, CBD significantly inhibits the myeloperoxidase activity of neutrophils [130].

Recent studies evidenced that neutrophils adopt distinct functional phenotypes, namely N1 and N2, which have pro- and anti-inflammatory characteristics, respectively. The specific phenotype they assume depends on the cues in their microenvironment and is influenced by the expression of specific cell markers [131]. In a mouse model of bilateral renal ischemia–reperfusion injury, treatment with CBD demonstrated significant reno-protective effects. This was accompanied by a reduction in the proinflammatory N1 phenotype and a decrease in Th-17 cells. Remarkably, CBD treatment also led to the restoration of the anti-inflammatory N2 phenotype and T regulatory (Treg)17 cells [132].

Furthermore, CBD inhibits the expression of both COX-1 and COX-2 mRNA, enzymes responsible for the conversion of arachidonic acid to prostaglandins in activated human polymorphonuclear cells. Prostaglandins, specifically prostaglandin E2, increase the sensitivity of nociceptors to stimuli and are important mediators of pain, constituting another mechanism by which CBD exerts its analgesic properties. The effects of CBD were compared with the reference NSAIDs showing high efficacy: CBD at a concentration of 1 µM downregulated the LPS-mediated increase of COX-1 and COX-2, similar to the effects of well-established COX inhibitors such as paracetamol and ibuprofen [133]. The CBD effects of neutrophils can be attributed to the inhibition of NF-κB and, potentially, the other transcription factors described above (Figure 4).

### 6.2. Effects on Lymphocytes

CBD affects humoral immune responses via a generalized suppressive effect on T cell functional activities. CBD attenuates the serum production of antigen-specific antibodies in mice and suppresses T cell proliferation and cytokine production, including IL-2, IL-4, and IFN-γ, both ex vivo and in vitro [134,135]. CBD significantly decreases the total number of CD4+ T cells in mice, as compared with vehicle treatment [136], and enhances apoptosis in three major subsets of normal splenic lymphocytes, including CD4+, CD8+, and B220+ [135]. A dose of 5 mg/kg/day causes lymphopenia by reducing B, T, T cytotoxic, and T helper lymphocytes [137,138]. Furthermore, when adult male rats were repeatedly treated with relatively low doses of CBD for 14 days, the total number of NKT cells increased, as well as the relative percentages of NKT and NK cells. Moreover, CBD regulates autoimmune memory T cells, defined as the Th17 phenotype, decreasing the production and release of IL-17, as well as that of IL-6 [139,140,141]. CBD leads to the upregulation of CD69 and lymphocyte-activation gene 3 (LAG3), regulatory molecules on CD4+CD25− accessory T cells, a recognized subtype of induced regulatory phenotype promoting anergy in activated T cells. Also, CBD treatment led to the upregulation of EGR2, which is a crucial inducer of T cell anergy. This upregulation was accompanied by increased levels of anergy-promoting genes such as IL-10 and STAT5. Additionally, CBD treatment promoted cell cycle arrest. Moreover, CBD had a significant impact on CD19+ B cells, since it decreased the levels of major histocompatibility complex class II (MHCII), CD25, and CD69, indicating a reduction in their antigen-presenting capabilities and a decline in their pro-inflammatory functions. The observed effects on lymphocytes can be attributed to the inhibition of NF-κB, IFN3, AP-1, and NFAT [139,140,141] (Figure 4).

## 7. CBD Affects the Fibrotic Response

Fibrosis is an irreversible scarring process characterized by excessive collagen and extracellular matrix component deposition promoted by immune cells that produce and release chemokines/cytokines and growth factors that enhance cell proliferation, tissue remodeling, and dysfunction [73].

As mentioned, CBD affects numerous biological functions and downregulates proinflammatory and profibrotic cytokines. CBD can reduce fibrosis by downregulating intracellular ROS generation and lipid peroxidation [73,142], inhibiting profibrotic signaling often associated with IL-6 [143] and IL-1 production [144], and favoring the transition of CD133+ progenitor cells into myofibroblasts [145].

CBD administration reduces inflammation and fibrosis in different experimental disease models, like multiple sclerosis, diabetes, cardiac ischemic disease, myocarditis, allergic asthma, and liver steatosis, among others [89,98,105,146,147]. Therefore, CBD may represent a pharmacological tool for reducing the pathological effects of aberrant fibrosis during organ remodeling following injury.

## 8. CBD Regulates Apoptosis

Apoptosis is a regulated mechanism of cell death, important in phases of tissue and organ development, cell activity regulation, and response to cell damage [148]. CBD induced apoptosis in different cancer cell lines, mainly breast carcinoma, glioma, leukemia, thymoma, neuroblastoma, and prostate and colon cancer, by activating caspase-8, caspase-9, and caspase-3, cleavage of poly (ADP-ribose) polymerase, translocation to mitochondria of Bid, and increasing the generation of ROS [149,150,151,152,153].

In breast cancer, CBD-induced apoptosis is accompanied by down-regulation of the mammalian target of rapamycin (mTOR), which regulates cell proliferation and apoptosis, cell cycle, and localization of PPARγ in the nuclei and cytoplasmic of the tested cells [154].

Additionally, CBD leads to dysregulation of calcium homeostasis, mitochondrial Ca^2+^ overload, stable mitochondrial transition pore formation, loss of mitochondrial membrane potential, and release of cytochrome c [151,155,156].

CBD modulates kinase activities in cancer cell lines, including inhibition of membrane-bound and intracellular kinases, and induces mRNA expression of several dual-specificity and protein tyrosine phosphatases, resulting in their dephosphorylation [149]. After CBD treatment, in human leukemia, in thymocytes and EL-4 thymoma cells, apoptosis was accompanied by an increased production of ROS and activation of NAD(P)H oxidases NOX4 and p22phox, further supporting the role of ROS in CBD-induced apoptosis [150,151].

Contradictory evidence exists pertaining to the influence of CBD on the apoptosis of normal, rather than transformed, immune cells [149,156,157].

CBD significantly reduces the apoptosis rate in a model of acute myocardial ischemia-reperfusion in rabbits [158]. Also, pretreatment of mice with CBD (10 mg/kg) in a model of hepatic I/R injury causes a significant decrease in apoptotic bodies at 24 h of reperfusion [159]. Moreover, CBD attenuates DNA fragmentation and poly(ADP-ribose) polymerase (PARP) activity [159].

In a mouse model of type I diabetic cardiomyopathy, CBD treatment for 11 weeks significantly decreased caspase-3 cleavage, caspase 3/7 activity, chromatin fragmentation, PARP activity, and apoptosis [105].

In a study conducted by Hsin-Ying Wu and colleagues [160], the impact of CBD on monocytes was explored in two distinct conditions: freshly isolated cells and cells precultured for 72 h. Surprisingly, they found opposite effects. In freshly isolated cells, CBD induced apoptosis, while precultured cells remained insensitive to its effects. These experimental findings propose a potential link between monocyte apoptosis and the reported anti-inflammatory properties of CBD.

Although clinical trials that specifically address immunomodulation and its possible complications are still lacking, no augmented incidence of infections or severe forms of them have been reported after seven years of experience with Sativex [19] and Epidiolex [8], the FDA-approved CBD compounds [161,162].

## 9. CBD Effect on Ion Channels

### 9.1. Calcium Channels

Calcium plays a crucial role in regulating various cellular processes, including excitation–contraction coupling, secretion, and the activity of numerous enzymes and ion channels. Within cardiac muscle, two types of Ca^2+^ channels are responsible for transporting Ca^2+^ into the cells: the L-type (low threshold type) and T-type (transient-type) channels. The L-type calcium channel is present in all types of cardiac cells, while the T-type calcium channel is predominantly found in the pacemaker cells, atrial cells, and Purkinje cells [163].

CBD modulates the ryanodine-sensitive intracellular Ca^2+^ stores in neurons [22,164]. This effect has not been proven in myocardial cells, but CBD significantly depresses electrically induced Ca^2+^ transients, further suggesting that it inhibits Ca^2+^-induced Ca^2+^ release.

In the presence of CBD, resting levels of intracellular Ca^2+^ and cell length in ventricular myocytes remain unchanged, indicating that it does not disrupt Ca^2+^ homeostasis under resting conditions [165]. However, several studies have provided evidence that CBD (1–10 μM) significantly inhibits voltage-dependent L-type Ca^2+^ channels in cardiomyocytes by accelerating the inactivation of these channels [165,166,167]. This inhibition reduces Ca^2+^-induced/Ca^2+^-release from the sarcoplasmic reticulum during excitation–contraction coupling [165,166,167], resulting in a negative inotropic effect in rat ventricular myocytes. Additionally, CBD acts as an inhibitor of recombinant human CaV3 channels and native mouse T-type currents. It shifts the steady-state inactivation of these channels to more negative potentials, reducing the number of open channels upon cell depolarization. However, CBD does not affect channel activation or the decay of currents after opening, indicating a possible open channel block [168].

In pathological conditions with elevated extracellular K+ levels, CBD can influence the activation of the Na+/Ca^2+^ exchanger (NCX), releasing additional Ca^2+^ from the cytosol into the extracellular space. At the mitochondrial level, CBD has a minor impact on mitochondrial Ca^2+^ regulation [169]; however, during ischemia–reperfusion, CBD can restrict the entry of Ca^2+^ into the mitochondria by avoiding IP3-dependent Ca^2+^ liberation from the sarcoplasmic reticulum, therefore preventing further injury caused by the overflow of Ca^2+^ [170,171] (Figure 5).

### 9.2. Sodium Channels

Voltage-gated sodium channels (NaVs) are hetero-multimeric membrane proteins responsible for the rapid upstroke of the action potential (AP) in excitable cells, allowing an influx of Na+ ions (INa) down their concentration gradient. Of the nine human voltage-sensitive sodium channel isoforms, NaV1.4 and NaV1.5 are primarily expressed in skeletal and cardiac muscle cells [172,173,174].

CBD is a nonselective sodium-channel inhibitor and creates a steep average Hill slope, suggesting multiple interactions. It prevents the activation of sodium channels from rest while also stabilizing the inactivated states of these channels without altering the voltage dependence of activation [166,170].

At a concentration of 3.3 μM, CBD can inhibit about 90% of the sodium conductance. However, the remaining population of channels, unaffected by CBD, maintain their original voltage dependence of activation without any significant changes in the midpoint or apparent valence of activation. Consequently, exposure to CBD at this concentration prevents those channels from conducting, while leaving their voltage activation characteristics unaltered [170,173] (Figure 5).

### 9.3. Potassium Channels

Cardiac potassium (K+) channels can be classified into three main categories: voltage-gated channels (including Ito, IKur, IKr, and IKs), inward rectifier channels (such as IK1, IKAch, and IKATP), and background K+ currents (comprising TASK-1 and TWIK-1/2 channels). The expression levels of these channels vary across the heart, leading to regional differences in the action potential (AP) characteristics in the atria and ventricles, and across the myocardial wall [163].

CBD suppresses the delayed rectifier currents IKr and IKs, while it has less impact on the transient outward current Ito and inward rectifier IK1 [167,175]. In Purkinje fibers, CBD demonstrates a stronger reduction in the AP duration at half-maximal repolarization compared to near-complete repolarization. It also causes a slight decrease in the AP amplitude and its maximal upstroke velocity. However, CBD does not exert any significant effects on the membrane’s resting potential [74,167].

Moreover, CBD functions as a competitive inhibitor of the equilibrative nucleoside transporter (ENT), which hinders the cellular uptake of adenosine. This leads to an increase in the extracellular concentration of adenosine, facilitating the heightened activation of adenosine A1 receptors [74,176,177]. Consequently, the activation of these receptors triggers the opening of potassium K+-ATP channels, resulting in hyperpolarization of the cell membrane.

Thus, CBD prolongs repolarization, increasing the action potential duration [56,175]. Overall, low or high CBD concentrations can induce arrhythmic effects in rabbits, mice, and rats [74,167]. In opposition, CBD suppresses ischemia-induced ventricular arrhythmias and exerts cardioprotective effects [74,169].

In human clinical trials, in healthy individuals under normal physiological conditions, CBD administration poses a minimal or insignificant risk of causing proarrhythmic effects. This aligns with clinical observations, in which CBD administration did not result in significant QTc prolongation in patients. However, caution should be exercised when using CBD concurrently with drugs that influence cardiac repolarization or impair drug metabolism, as well as in certain pathophysiological situations, such as hypokalemia, channelopathies, diabetes mellitus, hypertrophic myocardiopathy, or heart failure, among others, CBD can have an additive effect, further increasing the proarrhythmic risk and the possible incidence of sudden cardiac death (Figure 5).

## 10. Conclusions

CBD is active in different cellular and physiological processes like inflammation, apoptosis, oxidative damage, and fibrosis. The molecular targets where CBD exerts its direct effects are TRPV1, 5-HT1A, PPARγ, and L-type Ca^2+^ channels, which are responsible for its anxiolytic effects, blocking the transduction of chronic pain by affecting sensory nociceptors, modifying intracellular concentrations of calcium and the resting potential of excitatory cells, and profoundly affecting the inflammatory response by increasing anti-inflammatory cytokines and decreasing pro-inflammatory cytokines, monocyte adhesion, and neutrophiles’ trans-endothelial migration, as well as reducing the expression of the adhesion molecules. Also, CBD possesses intrinsic antioxidant effects. Also, by indirect mechanisms, CBD enhances the concentration and actions of endocannabinoids, adenosine, and GABA.

Despite considerable headway in unraveling CBD’s pharmacology and therapeutic potential, several gaps in knowledge persist. It is clear that CBD interacts with various cellular mechanisms; however, the precise mechanism(s) of action remains elusive. Furthermore, our existing knowledge predominantly originates from preclinical and cellular models, thereby raising concerns regarding genetic and physiological variations between species or models that might influence outcomes. Factors such as dosage, diverse metabolic profiles, timing of administration, and interactions with concurrent medications can introduce modifications to a study drug’s real-world implications.

The endorsement of cannabis-based medications poses unique regulatory challenges due to the intricate nature of cannabis and cannabis derivatives’ regulation and legal restrictions in different countries. As the scientific understanding of CBD and its therapeutic potential continues to evolve, regulations are likely to adapt and become more accommodating if the benefits will outweigh the risks.

Overall, CBD has been shown to have a wide range of effects, ones mainly described as beneficial, becoming an attractive potential treatment for a variety of acute and chronic disorders that involve auto-immunity, inflammation, tissue repair processes, or augmented oxidative damage.

## Figures and Tables

**Figure 1 molecules-28-05980-f001:**
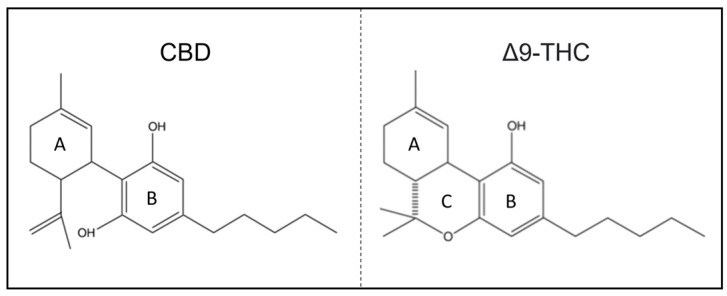
CBD and Δ9-THC chemical structures: cyclohexene ring (A–C) and aromatic ring (B).

**Figure 2 molecules-28-05980-f002:**
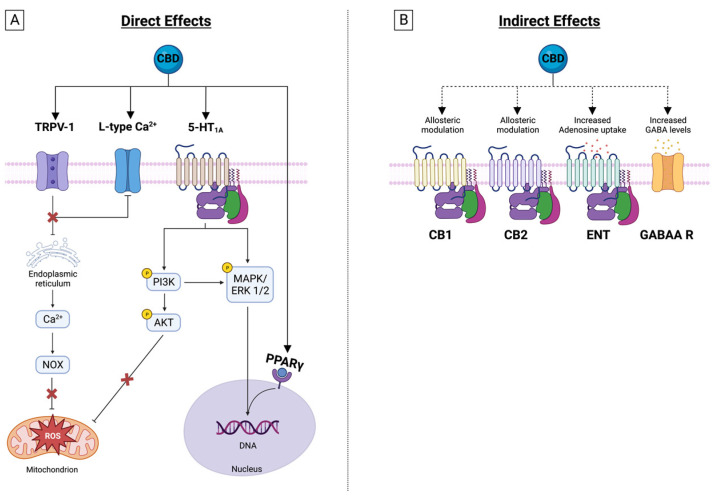
CBD receptors and intracellular signaling. (**A**) CBD interacts with several cell surface and nuclear receptors, antagonizing PI3K/AKT, MAPK/ERK, and JAK/STAT pathways. CBD inhibits through PPARγ receptor DNA transcription of proinflammatory mediators. Moreover, CBD modifies membrane and organelle calcium channels, altering intracellular signaling. (**B**) CBD exerts indirect effects on cannabidiol receptors and affects the uptake of adenosine and GABA, reinforcing their signaling.

**Figure 3 molecules-28-05980-f003:**
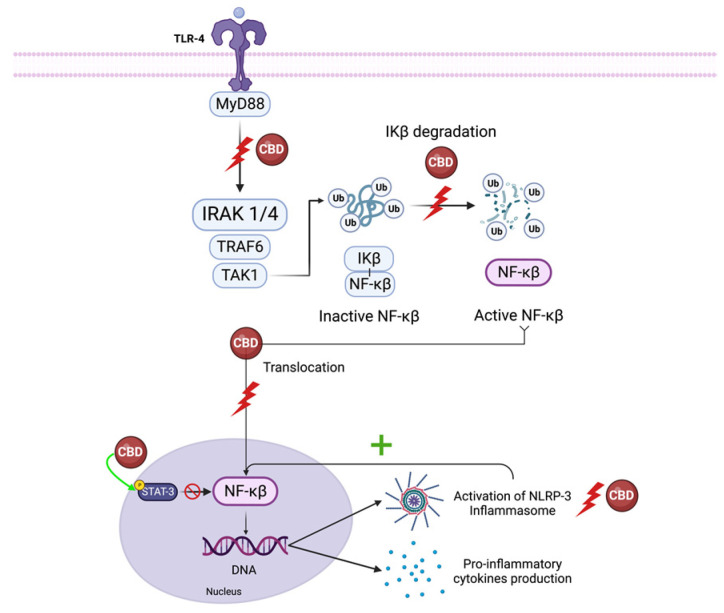
CBD effect on NF-κB pathway signaling. CBD decreases IRAK-1 and reverses the IkB degradation, ultimately reducing NF-κB translocation to the nucleus. Also, CBD suppresses NF-κB-mediated transcription by increasing anti-inflammatory STAT3 phosphorylation.

**Figure 4 molecules-28-05980-f004:**
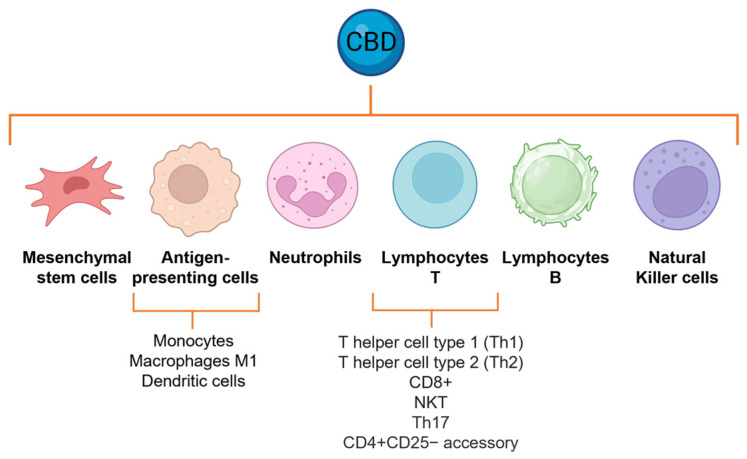
Immune cells regulated by CBD. CBD affects every stage of cellular inflammation, impacting the functionality of a wide range of immune cells by modifying a variety of mechanisms in favor of its immunosuppressive effect.

**Figure 5 molecules-28-05980-f005:**
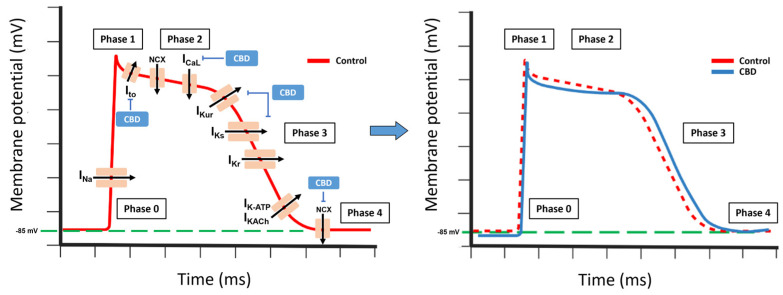
Effect of CBD ion channels. Myocardial action potential. Phase 0 (Depolarization); INa: Sodium current. Phase 1 (Sodium Channels Close). Phase 2 (Plateau); Ito: Transient outward potassium current; NCX: Sodium-calcium exchange; ICaL: Calcium current, calcium channel Type L. Phase 3 (Rapid Repolarization); IKur: potassium ultrarapid delayed rectifier current; IKs: Potassium slow-delayed rectifier; Ikr: Potassium rapid-activating delayed rectifier. Phase 4 (Resting Potential) IK-ATP: ATP-sensitive potassium channel; IKACh: muscarinic potassium channel.

## Data Availability

No datasets were generated or analyzed during the current study.

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
