# Peer review of "Molecular and Cellular Mechanisms of Action of Cannabidiol"

_molecules, 2023, doi:10.3390/molecules28165980_

Round 1

Reviewer 1 Report

The manuscript provides a comprehensive overview of Cannabis Sativa and its chemical compounds, with a focus on CBD. It covers various aspects, including pharmacology, pharmacokinetics, therapeutic uses, and potential adverse effects. The authors effectively present the current state of research on CBD's mechanisms of action and its potential as a therapeutic option. However, some sections may benefit from more concise explanations. The review highlights the need for further investigation and clarifies potential interactions with other medications.

1. What are the primary differences between phytocannabinoids and endocannabinoids in terms of their origin and physiological roles?

2. In the review, you mentioned both desired medicinal properties and prejudicial consequences of cannabis consumption. How do these effects vary depending on the method of consumption (e.g., smoking, edibles, oils)?

3. The review highlights the approval of a highly concentrated plant-based CBD preparation for certain medical conditions. Could you discuss the regulatory challenges in approving cannabis-based medications, and how do synthetic CBD formulations compare in terms of efficacy and safety?

4. CBD acts through multiple mechanisms, such as interaction with TRP channels, enhancement of endocannabinoid actions, and modulation of serotonin and GABA receptors. Are there specific medical conditions where one of these mechanisms plays a more significant role than others in CBD's therapeutic effects?

5. The review mentions that CBD has been extensively studied in cellular and animal models, but further investigation is required to confirm clinical effectiveness in humans. What are some of the challenges and limitations researchers face when translating findings from animal studies to human trials?

6. What are the current gaps in knowledge or areas of research that require more attention to better understand the pharmacology and therapeutic potential of CBD?

7. Given the widespread use of cannabis and its derivatives, what are the most pressing public health concerns related to cannabis consumption, and how can they be addressed through policy or public awareness campaigns?

Author Response

We thank the editor and the reviewers for the comments provided. In response to the reviewers' questions and comments, we have modified the manuscript and added new sections. We believe that with the modifications made in response to the reviewers have improved the overall content and quality of our manuscript. We hope that these will satisfy the editor and the reviewers as well. Here follows a detailed response to the reviewer’s comments.

Response to Reviewer 1

  1. What are the primary differences between phytocannabinoids and endocannabinoids in terms of their origin and physiological roles?

We thank the reviewer for raising this important question. The term ‘cannabinoid’ refers to the chemical substances with a characteristic C21 terpenophenolic structure that can interact with the cannabinoid receptors. There are two distinct groups of compounds that engage with the vertebrates’ endocannabinoid system (ECS): Endocannabinoids, which are endogenous lipid-based neurotransmitters within mammalian bodies, and phytocannabinoids, which are naturally occurring compounds found in the cannabis plant. From a chemical perspective, the endogenous ligands of endocannabinoids primarily consist of eicosanoid compounds synthesized in the lipid bilayer. They are rapidly synthesized and metabolized, resulting in a relatively short duration of action. They act “on demand” to uphold homeostasis and regulate various physiological processes, like pain, mood, appetite, sleep, and immune response. Phytocannabinoids differ significantly from endocannabinoids in structure. The duration of their effects can be prolonged depending on the method of consumption and the individual's metabolism. While CBD interacts with the ECS, it also exhibits other effects independent from its ECS activity. This review comprehensively outlines the relevant mechanisms, some of which overlap with endocannabinoids while others are different.

This information has now been added to the manuscript, and we believe it has improved content (Page 2, line 62-78).

  1. In the review, you mentioned both desired medicinal properties and prejudicial consequences of cannabis consumption. How do these effects vary depending on the method of consumption (e.g., smoking, edibles, oils)?

This question, posed by the reviewer, delves into another significant aspect. The way cannabis and its derivatives are consumed introduces variations in effects, metabolism, dosage, bodily distribution, and other pertinent factors. While these elements are indeed crucial, it is noteworthy that our current discourse primarily centers on CBD within the scope of this review. However, there is data on CBD that is important to take into consideration having to develop this compound into a drug. Controlled human studies to test CBD have predominantly utilized oral administration. However, CBD's oral bioavailability is limited, estimated to be as low as 6%. Existing data from orally administered CBD formulations indicate good tolerance, with only a few mild to moderate adverse events strongly related to dosage. Information on the differential rates of CBD absorption through various application routes, such as smoking, vaporization, oral soft gels, and oil drops, is scarce, inconsistent, and results in highly variable pharmacokinetic profiles.

Therefore, predicting the appropriate dose and route of administration to enhance desired effects and minimize adverse consequences becomes challenging in the absence of enough clinical data supporting the safety and efficacy of these compounds.

(Page 3, line 101 to 110).

  1. The review highlights the approval of a highly concentrated plant-based CBD preparation for certain medical conditions. Could you discuss the regulatory challenges in approving cannabis-based medications, and how do synthetic CBD formulations compare in terms of efficacy and safety?

We thank the reviewer for this additional important question regarding this topic. The approval of cannabis-based medications poses unique regulatory challenges due to the complex nature of cannabis as a plant, its active compounds, and its historical association with recreational use and legal restrictions.

Some of the key regulatory challenges in approving cannabis-based medications include: 1) Its legal status: it varies significantly across countries and even within different states or regions. 2) the lack of clinical data: there is still a relative shortage of robust clinical data supporting their safety and efficacy. 3) Dosing: Determining the appropriate dosage in cannabis-based medications is challenging since little has been published about that aspect. 4) Public Perception and Stigma: leading to challenges in obtaining support for research and approval of cannabis-based medications. 5) Regulatory Process: The existing regulatory frameworks were primarily designed for single-molecule drugs. Cannabis-based medications, with their complex mixtures of active compounds, may not fit neatly into these frameworks. This necessitates the development of new or adapted regulatory pathways tailored to cannabis-based treatments.

Despite extensive research investigating the pharmacokinetics of mixtures in different proportions of CBD-THC, there remains a notable scarcity of studies focusing solely on CBD.

CBD is generally regarded as a safe drug, with exceedingly rare reports of serious adverse events. However, natural CBD formulations may contain other phytocannabinoids at moderate concentrations, which could potentially result in side effects. Conversely, synthetic CBD production can yield a highly pure form with potentially fewer complications. Nonetheless, little has been published regarding its pharmacokinetic properties, particularly its bioavailability and accurate determination of effective doses.

As the scientific understanding of cannabis and its therapeutic potential continues to evolve, regulatory approaches are likely to adapt and become more accommodating to this unique class of medications. (Page 15 line 647 to 651).

  1. CBD acts through multiple mechanisms, such as interaction with TRP channels, enhancement of endocannabinoid actions, and modulation of serotonin and GABA receptors. Are there specific medical conditions where one of these mechanisms plays a more significant role than others in CBD's therapeutic effects?

We express our gratitude to the reviewer for highlighting this aspect. Some implications for the effects of CBD on these receptors have been discussed in the review paper.

Regarding the potential therapeutic effects of CBD across various diseases, research has indicated that TRPV1 receptors undergo desensitization, reducing pain perception and pro-inflammatory signals. It remains to be elucidated the specific effects and mechanisms of CBD on analgesia and chronic pain management, but TRPV-1 receptors may be involved. Also, CBD has shown positive effects in models of rheumatoid arthritis in rodents by activation of TRPV1 receptors. Moreover, activation of TRPV1 has been implicated in the pathophysiology of diseases such as cystitis, asthma, and hearing loss. On top of that, CBD may affect 5HT1A serotonin receptor signaling which mediates mood, depression, and anxiety, but it remains to be understood if CBD alone can provide these effects. Regarding GABA receptors, CBD could interfere in different pathophysiological brain processes such as anxiety and sleeping disorders and may reduce seizures because of GABA regulation. Also, GABAA have been implicated in the pathogenesis of Down Syndrome, Affective Disorders, Schizophrenia, and Autism

Finally, by regulating inflammation, oxidative damage and apoptosis, CBD has shown positive effects in animal models of autoimmune diseases, Alzheimer’s disease, and cancer among others. Aditionally, it has shown neuroprotective and cardioprotective effects on brain and cardiac ischemia-reperfusion injury animal models.

While certain implications regarding the effects of CBD on these receptors have been addressed, we are presently delving further into the exploration of these considerations. This is particularly pertinent as the causal relationship between these receptors and the aforementioned diseases is not yet firmly established.

  1. The review mentions that CBD has been extensively studied in cellular and animal models, but further investigation is required to confirm clinical effectiveness in humans. What are some of the challenges and limitations researchers face when translating findings from animal studies to human trials?

This is a recurrent question that arises when preclinical and cellular models are used to model human disease. As discussed before, regulatory processes can affect even more the application and translation of animal findings to human studies. Some common obstacles that arise to translating treatments from cellular and animal models into the clinic are genetic and physiological differences between the species or models that may affect the results. One of the primary challenges lies in the variability of conditions among the patients participating in clinical trials, which complicates the interpretation process. In contrast, cellular and animal models offer the advantage of easily controlled conditions leading to more straightforward results and interpretations. Furthermore, dosage and different metabolism, or time of treatment (pre-or post-), and interaction with other medications can modify the real world in a study drug. Finally, patients may have comorbidities, and it is necessary to consider possible safety issues for certain diseases. Even if the potential adverse reactions are not serious, they represent a limitation that should be addressed, and a cost-benefit analysis should be conducted. These points are now taken into consideration in the conclusion (Page 15 line 642 to 642).

  1. What are the current gaps in knowledge or areas of research that require more attention to better understand the pharmacology and therapeutic potential of CBD?

While significant progress has been made in understanding the pharmacology and therapeutic potential of CBD, there are still several gaps in knowledge and areas of research that require more attention to gain a comprehensive understanding.

To our understanding, the principal gaps in knowledge are its mechanisms of action and dosage.

Although we have a general understanding of CBD's interactions with several receptors, the notion that there is not a specific and unique receptor suggests that there exists a specific molecular mechanism underlying its various therapeutic effects and, up to date, that remain incompletely understood. Also, determining the optimal dosage of CBD for specific medical conditions is challenging due to the lack of well-controlled clinical trials with standardized dosing regimens for various indications in order to maximize therapeutic efficacy while minimizing the risk of adverse effects.

Addressing these gaps in knowledge will contribute to a more comprehensive understanding of CBD's pharmacology and therapeutic potential, enabling healthcare professionals to consider its use as a safe and effective treatment option for various medical conditions. These points are now taken into consideration in the conclusion (Page 15 line 639 to 646).

  1. Given the widespread use of cannabis and its derivatives, what are the most pressing public health concerns related to cannabis consumption, and how can they be addressed through policy or public awareness campaigns?

This is a very important and controversial point in the US and in most countries. The most pressing public health concerns related to cannabis consumption include:

Adolescent Use: Cannabis use during adolescence can have adverse effects on brain development, leading to impaired cognitive function and increased risk of mental health issues.

Driving Impairment: Cannabis consumption can impair cognitive and motor functions, leading to an increased risk of accidents and injuries while driving.

Addiction and Dependence

Risk for Vulnerable Populations: Certain groups, such as pregnant women, individuals with mental health conditions, and those with pre-existing medical conditions, may face higher risks associated with cannabis consumption.

Regulatory Frameworks: Implementing comprehensive and evidence-based regulatory frameworks for cannabis production, distribution, and consumption can help ensure product safety and protect public health.

By combining effective policy measures with targeted public awareness campaigns, societies can work towards addressing the public health concerns associated with cannabis consumption and promote responsible use while minimizing potential risks.

Well-designed public awareness campaigns can educate the public about the potential risks and benefits of cannabis use, target vulnerable populations, and promote responsible consumption. Also, improving access to evidence-based treatment and support services for individuals with cannabis use disorder can aid in addressing addiction and dependence issues.

Moreover, education programs in schools can help prevent adolescent cannabis use and provide students with accurate information about cannabis and its potential effects, driving education and enforcement could strengthening public awareness on the risks of impaired driving due to cannabis use and implementing effective enforcement measures can discourage driving under the influence of cannabis.

While we think that these points are as important as the understanding of the mechanisms of action of phytocannabinoids and CBD, this review focuses on the latter. Highly purified formulations of CBD are not the same as administration of cannabis sativa, therefore this consideration must be taken into account.

Author Response

We thank the editor and the reviewers for the comments provided. In response to the reviewers' questions and comments, we have modified the manuscript and added new sections. We believe that with the modifications made in response to the reviewers have improved the overall content and quality of our manuscript. We hope that these will satisfy the editor and the reviewers as well. Here follows a detailed response to the reviewer’s comments.

Response to Reviewer 2

*THE QUALITY OF IMAGES HAS BEEN ENHANCED TO IMPROVE THEIR VISUAL APPEAL.

-The CB1 receptor is mainly known for its psychotropic effects, on the contrary in recent years several literature data describe an important role for CB2 agonists (also synthetic molecules) which have and anti-proliferative and pro-apoptotic actions, as well as anti-apoptotic action, as well as anti-inflammatory effects. How can CBD affect CB2 signaling? What can be the benefits of using CBD rather than specific CB2 agonists?

We thank the reviewer for this comment, as the topic is very complicated. Studies have shown that CBD can influence the expression and density of CB2 receptors on cell membranes. It can upregulate or downregulate the number of CB2 receptors, thereby affecting the overall responsiveness of cells to endocannabinoids.

Over time, scientists have developed and created new ligands that specifically target CB2 receptors (CB2R) while prioritizing high selectivity over cannabinoid receptor type 1 (CB1R) to avoid unwanted psychotropic effects associated with CB1R activation. Despite these efforts, the clinical outcomes using these CB2R ligands have shown limited effectiveness. The therapeutic conclusions related to CB2 agonists have predominantly relied on the effects of nonselective and nonspecific first-generation ligands (such as JWH133, AM1241, AM630, and others). However, these conclusions have not been validated using more selective ligands. In addition, several studies have inferred a role for CB2 receptors (CB2R) in behavioral or other central nervous system (CNS)-mediated effects based on the use of SR144528 as an antagonist. However, it is important to note that SR144528 has very poor brain penetrance, meaning it does not effectively cross the blood-brain barrier and reach the brain in significant amounts.

The use of CBD over specific CB2 agonists offers several advantages. CBD's non-selective activity allows it to interact with various receptors and signaling pathways in the body, providing a broader range of therapeutic effects compared to specific CB2 agonists. Moreover, CBD's anti-inflammatory effects extend beyond CB2 receptor activation.

While specific CB2 agonists may have their advantages in certain cases, such as in targeted immune modulation, using CBD offers a more comprehensive and versatile approach to address multiple health conditions with fewer potential drawbacks. As research on CBD and its effects continues to expand, its therapeutic potential and utility in various medical contexts are becoming increasingly evident.

These data and considerations have been added to the review (Page 4 line 137 to 140; 151 to 160)
